# Sex Determination and Differentiation in Teleost: Roles of Genetics, Environment, and Brain

**DOI:** 10.3390/biology10100973

**Published:** 2021-09-27

**Authors:** Preetha Rajendiran, Faizul Jaafar, Sonika Kar, Chenichery Sudhakumari, Balasubramanian Senthilkumaran, Ishwar S. Parhar

**Affiliations:** 1Brain Research Institute Monash Sunway (BRIMS), Jeffrey Cheah School of Medicine and Health Science, Monash University Malaysia, Bandar Sunway 47500, Malaysia; Preetha.Rajendiran@monash.edu (P.R.); Faizul.Jaafar@monash.edu (F.J.); 2Department of Animal Biology, School of Life Sciences, University of Hyderabad, P O Central University, Hyderabad 500046, Telangana, India; nehadoc.1993@gmail.com (S.K.); sudhacc@yahoo.com (C.S.); bsksl@uohyd.ac.in (B.S.)

**Keywords:** sex determination, sex differentiation, bipotential gonad, brain

## Abstract

**Simple Summary:**

The fate of the gonad in teleost is influenced by various factors, including genetics and external factors. Several species-specific genes and environmental factors involved in sex determination and differentiation have been identified in teleost. In addition, these factors are species-specific. At the brain level, suppression of key molecule of the hypothalamus–gonadal axis affects sex determination. At the same time, pituitary hormones are required for regulating sex differentiation. However, the role of the brain during sex determination and differentiation remains elusive. In this review, we have gathered and discussed the findings on the role of prominent genes, environmental factors, and the brain in regulating sex determination and differentiation of teleost.

**Abstract:**

The fish reproductive system is a complex biological system. Nonetheless, reproductive organ development is conserved, which starts with sex determination and then sex differentiation. The sex of a teleost is determined and differentiated from bipotential primordium by genetics, environmental factors, or both. These two processes are species-specific. There are several prominent genes and environmental factors involved during sex determination and differentiation. At the cellular level, most of the sex-determining genes suppress the female pathway. For environmental factors, there are temperature, density, hypoxia, pH, and social interaction. Once the sexual fate is determined, sex differentiation takes over the gonadal developmental process. Environmental factors involve activation and suppression of various male and female pathways depending on the sexual fate. Alongside these factors, the role of the brain during sex determination and differentiation remains elusive. Nonetheless, GnRH III knockout has promoted a male sex-biased population, which shows brain involvement during sex determination. During sex differentiation, LH and FSH might not affect the gonadal differentiation, but are required for regulating sex differentiation. This review discusses the role of prominent genes, environmental factors, and the brain in sex determination and differentiation across a few teleost species.

## 1. Introduction

Teleost have the most diverse reproductive system and reproductive strategies among vertebrate species. Therefore, teleost is an exciting group of organisms to investigate the evolution of sex determination and sex differentiation due to their vast range of reproductive systems, from hermaphrodites to gonochoristic and plasticity of adult sex change [1]. These characteristics allow an opportunity to analyse the differences in the structure and expression of genes responsible for sex determination and sex differentiation.

Gonadal development in teleost is complex and elusive due to phenotypic plasticity. Nonetheless, the basic mechanism of gonadal development remains similar across teleost (Figure 1) [2]. The testis and the ovary originate from a bipotential gonadal primordium [3], which consists of a few primordial germ cells and somatic cells (SCs). The development of gonads starts with sex determination. Sex determination acts as a master switch to bipotential gonadal primordium and activates the differentiation pathway. Sex determination can be genetic sex determination (GSD), environment sex determination (ESD), or both, depending on the species [3]. In teleost, a more convoluted system can include various sex chromosomes, numerous gene loci, and diverse sex determination systems. During sex determination, several prominent genetic and environmental factors are involved. For example, there are genes such as *amhr2* [4], *amhy* [5], *dmrt1* [6], *dmy* [7], *gdf6Y* [8], *gsdf* [9], *gsdf^Y^* [10], and *sdY* [11]. At the cellular level, most sex-determining genes are involved in the suppression of the female pathway.

Once the fate of a bipotential gonadal primordium is determined into a testis or an ovary, sex differentiation takes over the gonadal development process. In teleost, sex differentiation is labile and influenced by genes, hormones, and extrinsic factors throughout gonadal development [12]. Several players are needed to achieve the end goal, i.e., the maturation of a male or a female fish. Genes that are prominently involved in sex differentiation include *amh* [13], *amhr2* [14], *amhy* [15], *dmrt1* [16,17], *cyp19a* [18,19], *figla* [20], *gsdf* [21], and *sox9* [22]. Sex differentiation is the product of a combative relationship between genes involved in the development and maturation of testis and ovary [23,24]. The environmental factors here are temperature [25], pH [26], population density [27], oxygen concentration [28], and social status [29]. However, at this stage, the involvement of environmental factors is more on maintaining the activation and suppression of various male and female pathways depending on the sexual fate.

The brain is known to play an essential role in regulating many bodily systems in an organism, including the reproductive system. In teleost, the brain is sexually differentiated into a male and female brain, but has the ability to change and adapt [30]. Nonetheless, the role of the brain during sex determination and differentiation is poorly understood until today. A previous study has shown that gonadotropin-releasing hormone III (GnRH III) knockout promotes a male sex-biased population [31]. This finding shows there is an involvement of the brain during sex determination. During sex differentiation, luteinizing hormone (LH) and follicle-stimulating hormone (FSH) hormones have no effect on the testis or ovary differentiation [32]. However, both hormones are required for gonadal hormone synthesis, which regulates sex differentiation. Therefore, this shows the involvement of the brain during sex determination and differentiation. In this review, we collectively discuss the role of prominent genes involved in sex determination and differentiation across a few fish species. Furthermore, the discussion continues as to how the internal and external environmental factors and the brain control sex determination and differentiation in teleost.

## 2. Regulation of Sex Determination

### 2.1. Genetics

Genetic regulation of sex determination in teleost has been studied extensively. Genetic sex determination (GSD) can be classified into a single gene (Japanese medaka, *Oryzias latipes*) or polygenic (zebrafish, *Danio rerio*) sex determination. GSD in teleost is not well conserved and is species-specific. Several sex-determining genes have been identified in teleost; their description is presented in the subsequent paragraph and Table 1.

#### 2.1.1. The Anti-Müllerian Hormone Receptor Type 2 (*amhr2*)

In mammals, the binding of the anti-Müllerian hormone (Amh) to its receptor (anti-Müllerian hormone receptor type 2; Amhr2) activates subsequent pathways that prevent the development of the Müllerian ducts into the uterus and fallopian tubes [39]. The *amhr2* gene was classified as a sex determination gene, first in the grass puffer [40]. The *cyp19a* promoter with SMAD4 binding sites is activated when the Amh binds to the Amhr2 [41].

#### 2.1.2. The Y-Linked Anti-Müllerian Hormone (*amhy*)

The Y-linked anti-Müllerian hormone (*amhy*) is a non-transcription factor protein and a member of the transforming growth factor-β (TGF-β) superfamily. The *amhy* gene, located downstream of *amh* on the Y chromosome, is a duplication of the autosomal *amh* gene [42]. As mentioned above, Amh is secreted to inhibit the female reproductive Müllerian duct formation [39,43]. As the Müllerian duct is absent in teleost [44], Amhy protein regulates the expression of *foxl2* and *cyp19a1a* mRNAs [45] and plays a critical role as a sex determination gene in male fishes. *amhy* was discovered as a sex determination gene in the Patagonian pejerrey (*Odontestes hatcheri*) [46] and, more recently, in the Nile tilapia [5], cobaltcap silverside [33], Northern pike [34], and rockfish [35].

#### 2.1.3. The Doublesex and Mab-3 Related Transcription Factor (*dmrt1*)

The doublesex and mab-3 related transcription factor (*dmrt*) gene family is a well-conserved gene classified by a DNA-binding region known as the DM domain. In most mammals, the *dmrt* gene is present in more than one variant [47]. There are eight functional *dmrt* genes in mammals, *dmrt1-8*. Meanwhile, there are six *dmrt* gene variants commonly found in teleost, *dmrt1-6.* However, not all the variants of this gene are involved in gonadal development [47]. The *dmrt1* gene has been identified as a candidate for sex determination in the spotted scat [6] and Chinese tongue sole [36], and is the only gene associated with male sex determination in teleost.

#### 2.1.4. The DM-domain on the Y-Chromosome (*dmy/dmrt1by*)

The DM-domain gene on the Y-chromosome (*dmy*) is the first sex-determining gene found in teleost. Furthermore, *dmy* is a species-specific and primary gene responsible for the sex determination of Japanese medaka [7,48]. Japanese medaka is a gonochoristic fish that develops into a distinct female with ovaries and a distinct male with testes. In addition, male Japanese medaka possesses heteromorphic sex chromosomes, XY. In contrast, female Japanese medaka possesses homomorphic chromosomes (XX). *dmy*, also known as *dmrt1by* gene, is a male sex-determining gene in Japanese medaka. This gene is identified as a duplicate of the autosomal *dmrt1* gene found in the sex-determining region of the Y-chromosome. Similarly, the *SRY/Sry* gene, a sex determination gene in mammals, is derived from duplication of autosomal *Sox* gene. With this similarity, the researcher strongly suggests that the *dmy* gene has an equivalent function to the *SRY/Sry* gene of mammals as a sex-determining gene in Japanese medaka [7].

#### 2.1.5. Growth Differentiation Factor 6 on the Y-Chromosome (*gdf6Y*)

Growth differentiation factor 6 on the Y-chromosome (*gdf6Y*) gene is encoded for gdf6Y protein, one of the TGF-β family. gdf6Y is a secreted ligand involved in the growth and differentiation of developing embryos [8]. The *gdf6Y* gene is located in the male-specific region of the Y-chromosome. To date, the *gdf6Y* gene is classified as sex-determining gene only for the turquoise killifish from genome sequencing [8]. Nonetheless, the function of the *gdf6Y* gene in sex determination requires further analysis.

#### 2.1.6. The Gonadal Soma Derived Factor (*gsdf*) and the Gonadal Soma Derived Factor on the Y-Chromosome (*gsdf^Y^*)

The gonadal soma derived factor (*gsdf*) gene encodes for Gsdf protein, a member of the TGF-β superfamily. The *gsdf* gene is postulated to be an ancestral gene responsible for male sex determination [49]. Additionally, *gsdf* also plays a critical role in regulating testicular germ cell proliferation and differentiation [50,51,52]. In teleost, the location of the *gsdf* gene on the chromosome is species-specific. A previous study showed that, within the medaka family (*Oryzias)*, *gsdf* is present on either the autosomal chromosome, as in the Japanese medaka, or on the Y-chromosome, as in the Philippine medaka [10,53]. Interestingly, the *dmy* gene is absent in the Philippine medaka. Identification of sex determination in Philippine has revealed that the function of the *dmy* gene is replaced by the *gsdf* gene on the Y-chromosome (*gsdf^Y)^*). Besides the Philippine medaka, the autosomal *gsdf* gene (chromosome 6) has also been classified as a sex determination gene in the rainbow trout [9].

#### 2.1.7. Sexually Dimorphic on the Y-Chromosome (*sdY*)

Similar to other sex-determining genes in a teleost, sexually dimorphic on the Y-chromosome (*sdY*) is a species-specific sex determination gene. *sdY* has been classified as the sex-determining gene in most salmonid species (Salmonidae family), including rainbow trout [37], brown trout [11], Atlantic salmon [11], and Arctic charr [38]. The *sdY* gene is a duplication of interferon related factor 9 (*irf9*) gene which encodes Irf9 protein that is involved in the regulation of the immune system [54].

#### 2.1.8. SRY-Related HMG Box 3 on the Y-Chromosome (*sox3^Y^*)

*SRY* gene on the Y chromosome is the master initiator of testicular development in mammals [55,56], which has evolved from the X chromosomal *sox3* gene duplication [57]. Hence, *sox3*, identical to the Y-linked *SRY* gene, can substitute for its function, which was demonstrated upon induction of *sox3^Y^* in the gonads of transgenic XX mice wherein *sox3* was able to drive testes development in the absence of *Sry* [58]. In teleost, to generate a novel Y chromosome, the *sox3^Y^* gene can be independently recruited where *sox3* can activate downstream *gsdf* gene (a critical factor in fish male differentiation pathway) function [59]. Thus, *sox3^Y^* evolved as one of the sex-determining genes in a medaka-related species.

*sox3^Y^*, alone or with other transcription factors, can drive the male pathway directly or through steroidogenic enzyme regulation. In this context, *sox3^Y^* is vital in catfish during the late stages of gonadal development and seasonal maturation. Additionally, Sox3 emerged as a transcriptional activator of *11β-hsd* (a steroidogenic enzyme gene) by binding to specific promoter motifs [60]. Other essential functions of *sox3^Y^* (majorly expressed in developing gonads and in the brain) correspond to the formation of the hypothalamus–hypophyseal axis and neuronal differentiation [61]. Additionally, *sox3^Y^* acts as an apoptosis suppressor in ovary development, requiring follicle development and fecundity in zebrafish [62]. Although few reports in the teleost indicate *sox3^Y^* as a master sex-determining gene, further studies are needed across the species to conclude *sox3^Y^* as a significant player.

### 2.2. Environment

Environmental sex determination is well described in reptiles, particularly in crocodiles and turtles, while it remains elusive in teleost [63]. Several environmental factors are involved in sex determination, including temperature, density, hypoxia, pH, and social interactions [64]. Theoretically, the involvement of environmental factors in sex determination occurs before the critical period of sex differentiation. The exact mechanism on how environmental factors regulate sex determination remains unexplored. Nonetheless, previous studies have shown the three possibilities on how environmental factors regulate sex determination [64]; (i) environmental factors synergistically interact with genetic sex determination or (ii) override genetic sex determination, or (iii) primarily and independently regulate the sex determination.

#### 2.2.1. Temperature

Amongst the factors mentioned above, temperature is the prominent environmental factor in the sex determination of teleost [65]. The exact mechanism of temperature sex determination remains elusive. However, it is postulated that there are two ways in which temperature influences sex determination [65]. Firstly, coexisting with genetic sex determination. Previous studies have shown that high temperature upregulates the sex-determining gene, *dmrt1*, and causes a male sex-biased population [66,67]. Secondly, temperature acts independently, wherein it causes epigenetic changes to the gene required for sex differentiation. High temperature causes hypermethylation of *cyp191a* promoter and suppresses its expression [68,69], thus resulting in a male sex-biased population.

#### 2.2.2. pH

Apart from temperature, the pH of the water is also involved in the sex determination of teleost, mainly in the Cichlidae family. Previous studies have shown that pH determines the sexual development of *Apistogramma caetei* [26], *Pelvicachromis pulcher* [70], *Pelvicachromis subocellatus*, and *Pelvicachromis taeniatus* [71]. It has been shown that acidic water (pH < 7) results in a male monosex population or male sex-biased population [70,72], while neutral or basic water (pH ≥ 7) results in a female sex-biased population. Nonetheless, the exact molecular mechanism of how pH regulates sex determination remains unknown.

#### 2.2.3. Density and Hypoxia

Another interesting fact about teleost is that different population densities can influence sex determination. The catch from the wild is often a female fish [73,74], while fish in captivity, that grow in limited space and under high-density, results in a male sex-biased population. Hypoxia, a condition where the level of dissolved oxygen is low, is associated with the density of fish in captivity [75]. A high density of fish in captivity often results in a low level of dissolved oxygen. Undifferentiated gonads of zebrafish under a low level of dissolved oxygen result in a male sex-biased population compared to an average level of dissolved oxygen [28]. At the molecular level, both density and hypoxia activate the stress axis, or the hypothalamus–pituitary–adrenal axis, and upregulates the expression of cortisone. The conversion of cortisone is mediated by the 11β-HSD enzyme, which participates in androgen pathways specifically in the final step of 11-oxygenated androgens synthesis [76]. Thus, an increased level of cortisol results in an increased level of 11-ketotestosterone (11-KT), which induces male sex development.

#### 2.2.4. Social Interactions

Many teleost exhibit juvenile hermaphrodite or bipotential gonads. During the developmental stage, larger fry is often associated with masculinity, while smaller fry differentiates to females [77]. A contradicting study has shown that fish captured from the wild do not correlate between size and sex determination [78]. In contrast, fry in captivity with aggressive growth result in masculinisation and become male [79]. The precise mechanism of this phenomenon remains elusive.

### 2.3. Brain

The brain plays a vital role in regulating many systems, including the reproductive system. In mammals, particularly humans, the brain is sexually differentiated between males and females [30], while in a teleost the brain is also sexually differentiated, but has the ability to change and adapt. It is well understood that the brain regulates the reproductive system through the classical axis, the hypothalamus–pituitary–gonadal (HPG) axis. However, previous studies have been centred around identifying novel sex-determining genes in both gonads, testes and ovaries [3]. Several questions arise from this issue (Figure 2), (i) is the brain involved in sex determination? If yes, what is/are the mechanism(s) involved? (ii) Does sex determination of males or females occur in the gonads first, followed by brain sexual differentiation, or vice versa?

In general, the regulation of the HPG axis starts with the secretion of the key hypothalamic hormone, the Gonadotropin-releasing hormone (GnRH), a decapeptide secreted from the preoptic area of the hypothalamus into the hypophyseal portal system [80]. Once in the pituitary, GnRH stimulates the gonadotrophs of the anterior pituitary to secrete the gonadotrophin hormone, including FSH and LH. Both hormones are responsible for regulating testicular (spermiation) and ovary (ovulation) function by stimulating the synthesis of androgen and oestrogen, respectively [81]. Apart from regulating gonadal differentiation and function, both LH and FSH act as positive and negative regulators of the HPG axis. Previously, studies have shown that GnRH neurones do not express the oestrogen receptor α (Erα) or the androgen receptor (AR) [82]. Thus, the sex steroid feedback is relayed to the GnRH cells by regulating the upstream regulator of GnRH neurones [83]. Nonetheless, a recent study shows that GnRH neurons in Nile tilapia express ER [84]. In most teleost, there are three variants of GnRH (GnRH I-III), and some have only two, which are GnRH II and GnRH III [85]. In a recent finding, *GnRH III* knockout (*GnRH III* ^−/−^) in zebrafish has resulted in a male sex-biased population [31].

Furthermore, the absence of GnRH III in zebrafish upregulates the expression of genes involved in male gonad development such as *sox9a*, *amh*, and *cyp11*. In addition, inhibition of GnRH III suppresses the proliferation of primordial germ cells, which is one of the factors for male gonad development [86]. Zebrafish is classified as gonochoristic teleost, in which adult zebrafish appear as male or female but exhibit juvenile hermaphroditism. Therefore, the authors postulated that, by default, a zebrafish gonad is bound to become an ovary. Nonetheless, this theory would require further investigation as to how the brain regulates sex determination in teleost. Furthermore, gonadotropin-inhibitory hormone (GnIH) and kisspeptin exhibit inhibitory and stimulatory effect on GnRH, respectively [87,88]. Therefore, it can be speculated that these two molecules might be involved in sex determination through GnRH regulation, which remains unknown. Therefore, this could be another potential area of study to discover the role of GnIH and kisspeptin in sex determination.

## 3. Regulation of Sex Differentiation

Sex differentiation is the continuation process from sex determination. Similar to sex determination, sex differentiation involves a complex mechanism regulated by a single factor or interaction between multiple factors, including genetics, environment, and brain. In this section, we collectively describe the prominent genes and environmental factors involved in sex differentiation (Figure 3). Furthermore, the role of the brain during sex differentiation is also discussed in the subsequent paragraphs.

### 3.1. Genetics

#### 3.1.1. The Anti-Mullerian Hormone (*amh*) and Amh Receptor 2 (*amhr2*)

The Anti-Müllerian hormone (*amh*) is involved in a couple of processes in both sexes, namely male sex differentiation and female follicular development [89,90]. However, in some teleost, a negative relationship between *amh* and aromatase expression has been noticed during sex differentiation [13]. In zebrafish, high levels of *amh* accompany low levels of *cyp19a*, which suggests *amh* as a potential down regulator of *cyp19a*. The downregulation of the *cyp19a* might result in premature ovary-to-testis transformation [91].

In zebrafish, the Müllerian duct and *amh receptor 2* (*amhr2*) gene is absent, but it still retains *amh*. The absence of *amh* alleles in zebrafish results in female-bias ratios [92]. The mutant adult zebrafish have large testes where 50% of them have immature oocytes. It shows that *amh* controls male germ cell production and prevents the development or survival of oocytes [92]. Mutant males, compared to wild-type males, are less operational to stimulate wild-type females to lay eggs. Thus, *amh* is also crucial for male mating efficacy. Mutant females form sperm ducts, and some produce offspring. The young female mutants also lay a few fertile eggs, which infers functional sex ducts. However, as they age, they become sterile, which means for continuous fertility, *amh* is needed. The older ones yet have huge but sterile ovaries with collected non-vitellogenic follicles. Hence, *amh* is not vital for the growth of the reproductive ducts or the gamete formation initiation in zebrafish. Nonetheless, *amh* is essential for follicle proliferation and maturation and sustain fertility in males and females [92]. A study was conducted on Nile tilapia in 2015 to examine the role of *amhy* and *amhr2* in sex determination [5]. They found that overexpression of *amhy* and the knockout of *amhr2* in the XX Nile tilapia caused sex reversal. Therefore, they have hypothesized that both *amhy* and *amhr2* regulate aromatase expression to modulate sex determination [5].

#### 3.1.2. The Doublesex and Mab-3 Related Transcription Factor (*dmrt1*)

In Japanese medaka fish, despite having the *dmy/dmrt1 gene*, a copy of *dmrt1* in the sex chromosome, there is also an autosomal *dmrt* gene. There are four *dmrt* genes found in the Japanese medaka fish, which are *dmrt1*–*4* [96]. Amongst these four *dmrt* genes, *dmrt1* is the only gene of this family responsible for the differentiation of germ cells into testes [97]. However, *dmrt 2*, *3*, and *4* are expressed significantly during early embryogenesis [98]. Transcriptome analysis in Nile tilapia during differentiation revealed that *dmrt1* involves testicular differentiation and development [99]. At the molecular level, dmrt1 works antagonistically with foxl2 in testicular development. Expression of *dmrt1* in Sertoli cells upregulates *sox9b*, which promotes the transcription of testicular genes [100]. Furthermore, the expression of *dmrt1* in Sertoli cells suppresses the *foxl2* and subsequently the *cyp19a* gene.

#### 3.1.3. Aromatase (*cyp19*)

The aromatase (*cyp19*) gene encodes the aromatase protein, an enzyme that converts androgens to various oestrogen forms, a female sex hormone [101]. Aromatase in the endoplasmic reticulum supports the production, processing, and transportation of proteins. This gene expression depends on oestrogen’s need by the brain, retina, pituitary, and ovary [102,103,104]. In teleost, aromatase exists in two isoforms; *cyp19a* and *cyp19**b* encodes for two proteins, P450aromA and P450aromB, respectively. These two proteins are structurally distinct but with almost identical catalytic activities [105]. Cyp19a is also known as Cyp19a1, Cyp19a1a, and ovarian aromatase [106]. Similarly, cyp19b is known as Cyp19a2, Cyp19a1b, and brain aromatase [106]. In this review, we standardize the nomenclature to using Cyp19a and Cyp19b. The *cyp19a* gene, expressed predominantly in the ovary, is located in linkage group 18. In contrast, the *cyp19b* gene, expressed strictly in the brain, is located in linkage group 25, as evident in the zebrafish [107]. The *cyp19a* expressed in the ovary is primarily involved in oestrogen synthesis in the follicles’ granulosa cells. However, several reports show *cyp19a* in theca-interstitial cells of previtellogenic ovaries and interstitial cells of the testes in a teleost [108,109]. The *cyp19a* and *cyp19b* mRNA expression is found mainly in the gonads and the brain, respectively. The *cyp19a* gene in the brain is debatable, as other reports in the teleost showed that the cerebellum had either traces or no aromatase activity [110,111].

External factors, such as temperature, also regulate aromatase. It has been observed in several fish species that exposure to higher water temperature during pre-gonadal sex differentiation (early developmental stages) caused masculinization of the fish [105]. This temperature rise caused a couple of genotypic females to fail to differentiate into a complete phenotypic female while showing that elevated temperature downregulates aromatase. Treatment of juvenile zebrafish with a nonsteroidal aromatase inhibitor, fadrozole, or exposure to higher temperatures, resulted in a downregulation of *cyp19a* gene expression [112]. As teleost are poikilothermic, elevated temperatures of the surroundings can cause a spike in their growth curve. From this perspective, the high temperature makes several or more genotypic female fishes skip checkpoints during meiosis. For meiosis, it is more common to occur in females than males. Temperature can impact the rate of DNA methylation, which in turn could alter gene expression [105].

In the zebrafish, *cyp19a* expression is seen before and after gonadal differentiation. Low expression of *cyp19a* causes the juvenile ovary to disappear; germ and somatic cells form into testicular tissues. Exposure of the zebrafish to a higher concentration of oestrogens or oestrogenic compounds during development shows that the sex ratio changes drastically towards female dominance, with some of the fish having ovotestes [12]. In the zebrafish, knocking out the *cyp19a* gene but not the brain aromatase *cyp19b* gene results in all-male offspring [113]. The gonad’s fate, controlled by the expression of *cyp19a*, is controlled antagonistically by *dmrt1* and *foxl2* [16,114]. An experiment conducted on tilapia supports this hypothesis [16]. In that study, knocking out *cyp19a* and *foxl2* genes resulted in the females’ reversal of gonadal sex [16]. Concurrently, follicular cells neighbouring the degenerating oocytes express *dmrt1* and 11β-hydroxylase [16]. Similarly, during a protogynous change in the honeycomb grouper, the expression of *dmrt1* increased as the expression of *foxl2* decreased [115].

As mentioned previously, fadrozole is a drug that inhibits aromatase. Göppert and coworkers’ study induced secondary sex reversal in adult females of *Astatotilapia burtoni*, which showed a male-like phenotype following acute fadrozole treatment [116]. However, acute treatment with fadrozole toward male *A. burtoni* caused elevation of androgen levels and decreased oestrogen [117]. In addition, fadrozole treatment reduces aggressiveness in male *A. burtoni*. Prolonged treatment of fadrozole to female Nile tilapia with fully differentiated ovaries induces secondary sex reversal, in which the ovary completely transforms into fertile testes. Furthermore, serum levels of 17β-oestradiol (E2) are initially high and similar to untreated female tilapia, they then decrease significantly and reach similar levels as in male tilapia. After prolonged treatment of fadrozole, the low serum 11-KT in female tilapia increased and reached almost similar levels to male tilapia [118]. Fadrozole induces secondary sex reversal in the Nile tilapia and *A. burtoni*, but not in the common carp and goldfish. Treatment of fadrozole to these two fish species does not reduce serum E2 levels [116]. However, as the serum level of androgen was not measured in this study, the decisive mechanism of aromatase regulation on androgen and oestrogen in these fishes remains elusive.

#### 3.1.4. The Forkhead Box L2 (*foxl2*)

The forkhead box L2 (*f**oxl2*) gene is one of the *fox* gene family members that plays a considerable part in the female reproductive system, particularly in ovarian differentiation and oogenesis [119]. The *fox* gene is one of the essential ovary-specific genes which inhibit the growth of the ovary when suppressed. Mutation of this gene results in abnormal development of the ovaries [120]. Similar to the male reproductive system (the testes), the ovary consists of three major cell lineages: germ cells, granulosa, and theca cells [121]. During ovarian development, the proliferating germ cell exits the mitosis phase and starts the meiosis process, but is arrested at meiosis prophase I, transforming germ cells into oocytes. Furthermore, the granulosa cells act as supportive cells during ovarian development, enhancing germ cells’ growth. Lastly, theca and granulosa cells respond to the LH to produce steroid hormones such as oestrogen [119].

The *foxl2* gene belongs to a highly conserved gene family of transcription factors [122]. Furthermore, the *foxl2* gene has a conserved winged-helix domain, which binds DNA to a seven-base pair recognition motif on the 5’ promoter region. The DNA sequence of the *foxl2* gene in the pufferfish and zebrafish have similar sequences to the mammalian *foxl2* open reading frame, which is thought to be conserved orthologues [123]. Heterozygous mutations of the *foxl2* gene result in two syndromes, premature ovarian failure and a complete loss of ovarian expression. The Foxl2 protein plays an important role as a vital transcription factor during the initial ovarian differentiation and maintenance [119]. In addition, Foxl2 is also responsible for many developmental processes and cellular differentiation [122].

The study of *foxl2* gene function in the rainbow trout is of some interest. The rainbow trout is a distinctive model for this study for two reasons; (i) in evolution, the trout’s genome goes through a tetraploid and then a diploid phase; and (ii) despite sexual differentiation being controlled by genetic factors, which is the XX/XY mechanism, hormones can alter, resulting in the development of neo-XX males or neo-XY females that are fertile [124]. Interestingly, rainbow trout have two genetically independent *foxl2* gene paralogues, which are *foxl2a* and *foxl2b*. The expression of *foxl2a* in the trout is similar to the expression in mammals [123]. Meanwhile, the *foxl2b* gene is expressed sequentially after the *foxl2a* gene and is responsible for preventing and maintaining the ovary’s somatic cells from differentiating into a testis. In the oestrogen-treated neo-XY females, the somatic compartment of their ovaries expresses the *foxl2* gene. On the contrary, neo-XX males treated with androgen show suppression of *foxl2* gene expression. However, XX females treated with aromatase inhibitor show the expression of *foxl2* gene decrease exponentially, as in the tilapia [123].

Mutation of *foxl2* has been performed in the Nile tilapia to further understand the role of Foxl2 in sex determination [125]. Mutation of the *foxl2* gene in XX tilapia results in silencing of *foxl2* gene expression, and the gonads develop into testes. It is hypothesized that female to male sex reversal in the XX tilapia is due to the lack of *foxl2* expression. This reversal phenomenon is upregulated by male-dominant genes such as *sf1* (known as *ad4bp/sf-1* in the Nile tilapia), *gsdf*, and dmrt1. In contrast, genes observed in females, such as the *β-cat1*, *figla*, and *β-cat2*, are downregulated [125]. A similar phenomenon has also been observed in other organisms where knockdown of *foxl2* gene in XX tilapia [16], goat (XX) [126], and a double mutant zebrafish (*foxl2a*^−/−^/*foxl2b*^−/−^)[120] cause differentiation of female gonads into testes. In particular, sperms produced by sex-reversed males can fertilize eggs, and no difference is observed in fertilization rate than wild-type males [125].

*foxl2* influences the expression of the *sf1* gene, a gene mainly involved in gonadal development in males. Therefore, hypothetically, the presence of *foxl2* will downregulate the *sf1* gene expression and vice versa. A previous study has shown that the *sf1* gene is upregulated in *foxl2*^−/−^ XX gonads of the Nile tilapia [127]. However, more studies are required in other fishes to confirm this hypothesis, which will extend our understanding of sex-determining genes that specifically regulate gonadal development into one type of sex.

#### 3.1.5. Factor in the Germline Alpha (*figla*)

Factor in the germline alpha (*figla*) is a transcription factor with a basic helix–loop–helix structure [128]. In fishes, *figla* is a marker gene for ovary development and initial oocyte differentiation [129]. In the zebrafish, based on the expression levels of *figla* after post-fertilization day 26, individuals expressing low and high *figla* levels become males and females, respectively. *figla* is not a sex-determining factor, but females’ higher expression suggests that *figla* is vital for oocyte cyst breakdown, early folliculogenesis, and directing ovarian differentiation [130]. Similarly, in the Nile tilapia, *figla* is a gene-specific to females, expressed in the initial stage of primary oocytes; it is essential for both folliculogenesis and oogenesis. In the Chinese tongue sole, two genes encode *figla* (*figla_tv1* and *figla_tv2*) [129]. *figla_tv1* is expressed during the ovary’s cellular differentiation phase and persists into adulthood, and is responsible for ovary differentiation [131] and *figla_tv2* in spermatogenesis [129].

In *figla*-transgene male tilapia, the overexpression of *figla* relates to spermatogenesis impairment with elevated *hsd3b1* and 11-KT but no changes in *cyp17a1* and *StAR* in the Leydig cells [132]. Overproduction of 11-KT causes defects in spermatogenesis as elevated levels of androgen can cause early puberty, shrunken testes, and in some cases sterility in males [133]. StAR and P450ssc [134,135,136] are two crucial regulatory proteins for steroidogenesis in the gonads [137,138,139]. The upregulation of these two proteins suggests that *figla*_*tv2* might play a role in spermatogenesis as it regulates the synthesis and metabolism of steroid hormones in the gonads of pseudomales [129].

#### 3.1.6. The Gonadal Soma Derived Factor (*gsdf*)

In the Japanese medaka, the *gsdf* gene located on chromosome 12 is downstream of *dmy*. The *gsdf* gene, commonly expressed in the Japanese medaka, is found in the same types of cells as in the rainbow trout and Philippine medaka. The presence of *gsdf* in the absence of *dmy* leads to masculinization [53]. Furthermore, suppression of *gsdf* gene expression enhances the fish’s feminization; however, the expression of *dmy* remains unchanged. Following feminization, the beginning of the ovarian differentiation process takes place regardless of *dmy* expression. It has been hypothesized that *dmy* cannot replace the primary function of *gsdf*, which is to initiate testicular differentiation [53]. This study’s finding is supported by the fact that the mRNA of *gsdf* decreases by 28 folds in *dmy*^−/−^ XY gonads [53]. As the expression of the *gsdf* gene relates to the initiation of testicular differentiation, it is understandable that oestrogen suppresses it. However, the expression of *gsdf* gene is upregulated by androgen and higher temperature, factors that favour masculinization. The disruption of *gsdf* expression causes a cascade of changes in downstream pathways of *gsdf*, including the downregulation of *dmrt1* in adult gonads. Therefore, *gsdf* expands its function not only as an initiator, but also in maintaining the sex of the fish [53].

A recent study in the Nile tilapia has shown that deletion (*gsdf*^−/−^XY) or deficiency of *gsdf* gene sequence results in complete sex reversal [140]. Furthermore, in this study, the authors found that the expression of the *dmrt1* gene in the gonads remained unchanged during early developmental stages. In contrast, the sex reversal phenomenon of female phenotypic appearance does not occur in *gsdf*^−/−^XY male adult Japanese medaka [140]. However, another study has shown that suppressing *gsdf* gene expression in wild-type male Japanese medaka causes complete feminization in this fish [53]. Hence, the sex reversal process in fishes is multi-factorial and does not solely depend on a single gene [53].

*gsdf* gene regulation in the zebrafish works slightly differently than most other teleosts [95]. Studies have shown that the sex of either *gsdf*^−/−^ XX female or XY male zebrafish remains unchanged compared to the wild-type [95]. The *gsdf*^−/−^ XX female zebrafish are fertile for a short period. As the *gsdf*^−/−^ XX female zebrafish age, they become sterile due to accumulated non-vitellogenic follicles. The *gsdf*^−/−^ gene knockout in female zebrafish decrease the expression of the Vitellogenin *(vtg)* gene, which is involved in the synthesis of E2 [95,120]. Thus, lowering the level of E2 in *gsdf*^−/−^ XX female zebrafish subsequently impairs vitellogenesis, further increasing the number of non-vitellogenic follicles. On the other hand, *gsdf*^−/−^ gene knockout in male zebrafish show no impairment of fish fertility but develop large testes. Additionally, large testes cause upregulation of several genes including *vasa*, *smh*, *pcna*, *tp53*, *fshr*, and *casp3a* [95]. The *gsdf* gene is not located near any of the sex-linked loci in the zebrafish. Therefore, the *gsdf* gene is postulated as not a robust sex-determining gene in the zebrafish, and instead classified as a species-specific sex-determining gene [95].

#### 3.1.7. SRY-Related HMG Box 9 (sox9)

The SRY-related HMG box 9 (*sox9*) gene is encoded for Sox9 protein, a transcription factor that belongs to the HMG box family. Two variants can be found in teleost, *sox9a* and *sox9b.* Furthermore, the availability and function of both *sox9a* and *sox9b* are species-specific. In zebrafish, both *sox9a* and *sox9b* are differently expressed in Sertoli cells and oocytes, respectively [3]. During testicular development of zebrafish, *sox9a* is expressed highly in the bipotential gonad [141], and the expression of *sox9a* does not change during testicular differentiation. While in female zebrafish, the expression of *sox9b* is detected at a low level during the juvenile ovary. Throughout the ovarian stage, the expression of *sox9b* is dynamic, highly expressed at stages IB and II of oocytes and downregulated at stage III [141]. In Japanese medaka, only *sox9b* is expressed in both XX and XY supporting cells of the adult gonad, particularly in the oocytes and Sertoli cells, respectively [142]. *sox9b* is expressed at stage 36 of the Japanese medaka development stage prior to *dmy*, sex determination gene expression [143]. Interestingly, at the time of *dmy* gene expression, the levels of *sox9b* in both sexes is at the same level. Furthermore, *sox9b* is not involved in testis differentiation, but testis maintenance [144]. Suppression of *sox9b* in XY Japanese medaka does not show significant changes in the differentiation of PGCs to testicular tissue, whereas, in female Japanese medaka, the expression of *sox9b* in the granulosa is maintained before differentiation until diplotene oocytes exit from the germinal cradle [142]. Then, the expression of *sox9b* is replaced by *foxl2*.

### 3.2. Environment

#### 3.2.1. Sex Hormones

The endocrine system plays a vital role in sex differentiation [145]. Steroid hormones can induce phenotypic sex reversal in a teleost [146,147,148,149,150,151,152]. The production of oestrogen is related to ovarian differentiation. Meanwhile, 11-oxygenated androgens are involved in testicular differentiation. However, previous findings have shown that the balance of these two steroid hormones concludes gonadal sex differentiation rather than their absence or presence [153]. In the Chinook salmon, oestrogen synthesis inhibition using specific enzymes such as aromatase inhibitor causes a genetically female fish to undergo phenotypic masculinization [153]. Therefore, the absence of oestrogen is adequate to steer the gonad differentiation towards a testis. Oestrogen is needed for female sex differentiation, while the lack thereof results in male sex differentiation in an oestrogen-centric model [153].

Although testicular development might or might not require androgens, they must maintain the male phenotype, as the absence of them replaced with E2 minimizes the expression of a gene responsible for testicular differentiation, *dmrt1*. Therefore, due to the repression of the *cyp19a* gene, the male sex of the teleost is maintained. In the absence of oestrogen, it leads to impartial or complete functioning masculinization, which leads to the assumption that maintenance of the ovary requires constant production of oestrogen [153]. Therefore, oestrogen is an essential hormone for gonadal differentiation into ovaries to maintain this sexual form. Besides sex hormones, other studies have shown that cortisol is responsible for masculinization in a teleost [153]. This event occurs for two reasons; (i) a key enzyme in cortisol synthesis is also responsible for 11-oxygenated androgen synthesis, and this 11-oxygenated androgen favours testicular differentiation, and (ii) cortisol increases expression of the *cyp19a* gene. Again, this particular *cyp19a* gene experiences epigenetic inhibition when there is an increase in temperature resulting in masculinization of female teleost [153].

The use of hormones in aquaculture favours monosex culture to increase the growth rate in a short time [154,155]. The two common methods used to produce monosex fish are a direct approach, where fishes are treated with hormones that produce the wanted sex, and the indirect method, where the parent fish are treated with hormones that result in offspring being neomale, neofemale, or supermale, which produce same-sex larvae [150]. A wide range of natural E2, synthetic oestrogen, and synthetic androgen (17α-methyltestosterone) have been used to produce monosex fish. Both types of steroids are readily metabolized post-treatment [152,158,159].

Techniques of hormone treatment for sex reversal include injections, silastic implants, immersion, or hormones added to the feed [156]. Commercially in aquaculture, immersion and hormones in diet are the best practices due to their cost-effectiveness [157]. Compared to the immersion method, which requires knowing the type of hormone, the water temperature, and length of exposure, the feed method is viable and gives the fish the optimum dosage to induce and complete sex reversal [150,156].

A recent study in the European sea bass showed that genes related to ovarian differentiation such as *wisp1*, *cyp19a*, and *17β-hsd* and testicular differentiation such as *amh*, *dmrt1*, and *tesc* are downregulated after exposure to high temperature and E2 treatment. Suppression of all these genes results in the feminisation of the fish [158]. However, a contrasting report showed an increase in *cyp19a* gene expression following E2 treatment and suppression of *cyp19a* following exposure to high-temperature results in masculinisation [159]. The complexity of regulation of sex determination and sex differentiation in different environmental conditions remains unclear. Therefore, more studies are needed to fill the gap to understand the underlying mechanism of sex determination at the molecular level.

#### 3.2.2. Temperature

Some environmental factors could impact sex differentiation of teleost, including temperature, pH, and social interactions (refer to Figure 3). When the habitat temperature fluctuates, the biochemical pathways are affected, which results in biased sex differentiation [160]. Generally, in the thermosensitive teleost, temperature elevation induces testis development, which leads to a higher male population. On the other hand, low water temperature setting causes the development of ovaries [71]. This phenomenon occurs in a few species from the genus *Apistogramma* and *Dicentrarchus labrax L.*, a type of sea bass. The temperature shifts, however, is only crucial during the very initial process of sex differentiation. When a teleost is thermosensitive, it could be a hereditary trait. *Poeciliopsis lucida*, originating from Mexico, is a viviparous teleost with genetic polymorphism for sex determination and is influenced by the environment, in this case, temperature. In an experiment conducted using two strains of *P. lucida*, which are M61-31 and S68-4, the M61-31 strain produced a majority male offspring (169/187) at 30 °C; at 24 °C, the ratio was slightly skewed towards female offspring (250/395). However, the other strain, S68-4, produced equal proportions of males to females regardless of the temperature [161].

In some teleost species, an increase or decrease in temperature does not affect the sex of the offspring, such as *Cyprinodon variegatus* and Salmonid, *Coregonus hoyi* [162]. Thus, it shows that some teleost possess strong genetic sex determination, but can be mildly or not sensitive to the environment, such as the rainbow trout, Japanese medaka, and the common carp. On the other hand, species such as the sea bass are susceptible to a particular environmental factor, for example, a change in temperature or pH [71]. Finally, species such as the zebrafish are sensitive to multiple environmental factors; an increase in temperature, population density, and hypoxia can induce the masculinization of zebrafish [71]. It is known that in teleost with environmental sex determination, mostly temperature sex determination (TSD), their stress is facilitated by cortisol, which plays a significant role in activating the male pathway [163,164].

The effect of temperature on sex differentiation at the molecular level remains elusive. Nevertheless, a previous study has shown that the sex-determining gene, *cyp19a*, is involved in the TSD mechanism [165]. The *cyp19a* gene regulates ovarian differentiation of the *Odontesthes bonarensis* at a specific temperature [165]. The incubation of larvae of *Odontesthes bonariensis* at a masculinizing temperature suppressed the expression of the *cyp19a* gene [166,167,168]. Furthermore, the upstream regulator of the aromatase gene, *foxl2*, is also regulated by temperature during sexual differentiation in the *Paralichthys olivaceus*. In this species, the expression of the *foxl2* and the FSH receptor (*fshr*) genes were suppressed at high temperatures during sexual differentiation [169]. Foxl2 and FSH signalling are essential in regulating the transcription of the *cyp19a* gene during sex differentiation of TSD species [169]. A study conducted by Zhang and coworkers in 2018 showed that high temperature (29 °C) induces expression of *amhy*, a masculinisation signal, but suppresses ovarian differentiation pathway supported by *cyp19a.* Male pejerrey is seldom bred in low temperatures, and their ratio among XY individuals increases with elevated temperatures. This shows that *amhy*, a masculinising gene, is temperature-dependent for its expression [15].

#### 3.2.3. pH

In some environmental sex determination species, the pH of the water can influence sex differentiation. Acidic water induces masculinization in the *Poecilia melanogaster* [26], *Poecilia sphenops* [170], and *Pelvicachromis pulcher* [70]. The tambaqui (*Colossoma macropomum*), native species to Brazil, has a vast range of pH (4–7.2) as natural habitat [171], and in farming conditions, it sustains pH 4.0–8.0. Undifferentiated tambaqui were treated in either acidic (6.7) or alkaline (8.2) water for 45 days, pre-sex differentiation labile period; the control (pH 7.5) and the alkaline group produced a 1:1 sex ratio, whereas the pH 6.7 group resulted in a bias towards males (1.4:1); the bias could be a product of ionic stress [172].

*Pelvicachromis pulcher* is a small-sized cichlid found in west Africa [173,174]. The *P. pulcher* is sensitive to water pH during the developmental period. Acidic conditions produce a male-biased population when compared to neutral conditions [72]. Therefore, pH plays a role in the sex differentiation in *P. pulcher*, and it also alters phenotypic expression in males and females, modifying their propensity for aggression [70].

In a study conducted on rainbow trout, it was observed that low pH (5.6–6.0) leads to an increased level of plasma cortisol, a stress response to low pH [175]. Additionally, a high level of secreted T in the water could indicate a shift in the reproductive endocrinology of rainbow trout [175]. As mentioned previously, elevated temperatures induce stress in teleosts, causing a rise in cortisol [163,164]. Therefore, it could be postulated that low pH induces stress in teleosts resulting in high cortisol levels, causing a male bias population.

#### 3.2.4. Social Factors

Social factors also play a role in sex differentiation in hermaphroditic teleost. The social factor mainly regulates sex reversal in response to population density and the male to female ratio at a given period [176,177]. The precise mechanism of this effect is complex and not fully understood. Two different pathways produce *Thalassoma bifasciatum* males; (i) they can either mature as males naturally (primary males), or (ii) they mature as females and then undergo a sex reversal (secondary males). The high population density usually generates primary males. As for *Cichlasoma citrinellum*, the social factor affecting sex differentiation is the size of a juvenile where the relatively bigger fishes mature as males [178].

#### 3.2.5. Density

Population density can be a factor determining sex. Previous studies have shown that the high-density population (100 fishes/1.5L) of TU [179] and the AB [180] zebrafish strain produced a male-bias population, probably due to hypoxic conditions. During hypoxic conditions, the downregulation of genes responsible for synthesizing sex hormones and a surge of 11-KT and E2l in female zebrafish results in a male-biased population [181]. Furthermore, zebrafish embryos grown under hypoxic conditions show disruption in primordial germ cell migration, altered sex hormones concentration, increase in hypoxia-inducible factor-1 (HIF-1) signalling, which resulted in a male-biased population [28,186,187]. Given that hypoxia is a stress-inducing factor, it can instigate cortisol production, which prevents the development of ovaries while promoting masculinization [182]. Cortisol could also inhibit the expression of aromatase, subsequently raising the rate of apoptosis in the gonadal primordia or increase the synthesis of 11-KT and masculinize the fish [163,183,184,185,186].

#### 3.2.6. Hypoxia

Hypoxia and population density are interconnected as social factors. Theoretically, as the population’s density increases, the oxygen consumed by the fishes will naturally increase, resulting in lesser oxygen in the water. Therefore, a hypoxic state, which consequently causes stress among the fishes, could cause an increase in cortisol level, resulting in alteration of steroid levels in the fish. A study in the Amur sturgeon (*Acipenser schrenckii*) showed significantly high levels of cortisol post hypoxia stress [187]. It is possible that negative feedback of cortisol masculinises the fish during sex differentiation; cortisol inhibits the expression of aromatase, which then activates the pathway to develop male gonads [188].

### 3.3. Brain

The role of the brain during sex differentiation remains poorly understood. In the zebrafish, knockdown or suppression of pituitary hormones, LH and FSH, expression and secretion, does not significantly affect gonadal development [189]. However, knockdown and suppression of gonadotropins hormones only delayed the development of gonads, testis, and ovary [190,191]. Nonetheless, it is well known that LH and FSH regulate the expression of steroid hormones, including T and E2. In addition, both hormones, T and E2, are required for testicular and ovarian differentiation, respectively [192,193]. In addition, T and E2 have positive and negative feedback on the brain, thus regulating sexual function and behaviours [194,195].

In a study conducted on female orange-spotted grouper, *Epinephelus coioides*, *GnIH* mRNA levels in the hypothalamus fluctuate across different gonadal stages [196]. The expression of *GnIH* mRNA is low during primordial germ cells but increases during early differentiated ovary with primary oocytes, and decreases during vitellogenic stage. Similarly, the *GnIHR* mRNA expression levels in the pituitary vary throughout ovarian development. A previous study has shown GnIH peptide plays a role in the synthesis and secretion of T and E2 [197]. Furthermore, GnIH regulates the mRNA levels of *GnRH*, *lhβ*, and *fshβ* [198] and in the orange-spotted grouper, GnIH treatment decreases the expression of *GnRH* and *lhβ* mRNA. Besides GnIH, kisspeptin is also involved in sex differentiation of several fish species, including chub mackerel (*Scomber japonicus*) [199] and cinnamon clownfish (*Amphiprion melanopus*) [200]. Similar to GnIH, *kisspeptin* mRNA expression fluctuates across gonadal development and treatment of kisspeptin increases the expression of *GnRH*, *lhβ*, and *fshβ* [199]. Therefore, these studies indicate that GnIH and kisspeptin could play a role in sex differentiation.

Despite the fact there are no studies that show LH and FSH are responsible for the regulation of sex-differentiating genes, both FSH and LH are postulated to have a significant role in regulating sex differentiation. However, some questions related to this research remain unanswered; (i) Do LH and FSH interact with sex-differentiating gene to decide the fate of the gonads during sex differentiation? (ii) Is there a role for GnIH and kisspeptin in sex differentiation through the HPG axis, if so what is the mechanism involved? Answers to these questions would uncover new directions to enhance the understanding of sex differentiation, at the level of the brain.

## 4. Conclusions

The sex of a teleost, either male or female, is determined by the genome and other internal and external factors. Sex determination decides the fate of a bipotential primordium. Sex determination in teleost is controlled by genes, environment, or both. Genes involved in male sex determination include *amhr2*, *amhy*, *dmrt1*, *dmy*, *gdf6Y*, *gsdf*, and *sdY.* At the cellular level, most of the sex-determining genes are involved in suppressing the female pathway. Sex differentiation occurs after sex determination and involves the development of the gonad from the undifferentiated gonads. Sex differentiation is also dependent on genetic and environmental factors. Several genes that are prominently involved in sex differentiation include *amhr2*, *amhy*, *dmrt1*, *cyp19a*, *figla*, *gsdf*, and *sox9.* Environmental factors such as elevated temperature, a change in pH, oxygen concentration, population stocking density, and social status can determine the gender of fish. There are several intrinsic factors such as gonadal hormones (oestrogen and 11-KT) and stress hormone (cortisol) that, together with the synthesis of Cyp19a, change the fate of the gonad, i.e., sex reversal. Some genes are known to play a specific role in sex determination and differentiation. Despite the specific function of the gene, under certain extrinsic factors, the course of direction to form a specific gonad might be swayed and result in a different gonad. Furthermore, the role of the brain during sex determination and differentiation is still poorly understood today. *GnRH III* knockout promotes a male sex-biased population. While during sex differentiation, LH and FSH might not affect the testis or ovary differentiation, both are required for steroidal hormones synthesis, which also regulates sex differentiation. The mechanism of sex determination and sex differentiation remains elusive, particularly on the involvement of the brain. Therefore, more studies of brain and gonadal transcriptomic, together with top-down proteomics approaches and mass spectrometry, are needed to reveal new genes in the pathway of sex determination and differentiation. This will help to develop a sustainable ecosystem, particularly of endangered species, and for sustainable commercial culture.

## Figures and Tables

**Figure 1 biology-10-00973-f001:**
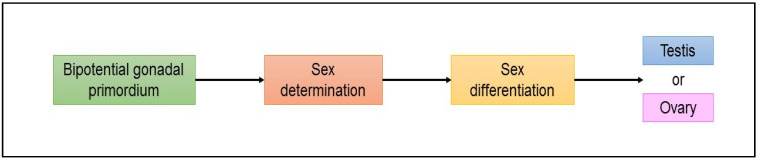
Basic pathway of gonadal development from bipotential gonadal primordium to become a testis or an ovary.

**Figure 2 biology-10-00973-f002:**
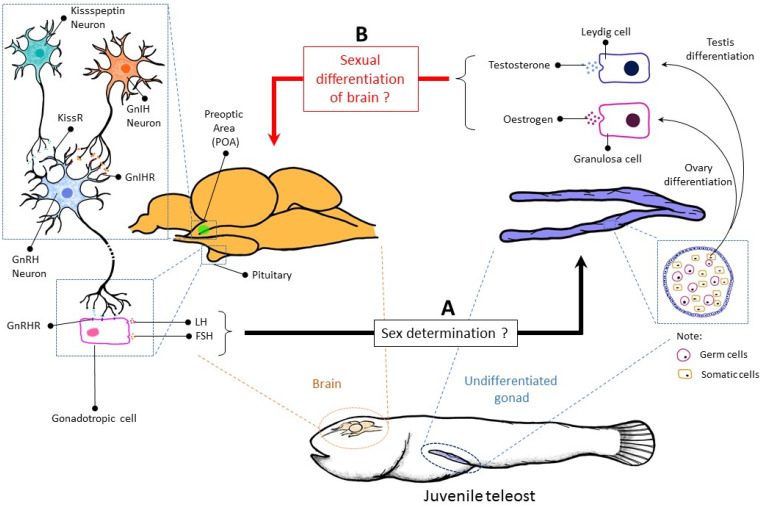
The schematic diagram of the uncertain pathway in the gonadal sex determination and sexual differentiation of the brain. (**A**) The role of the brain during sex determination remains elusive. There is only one study in zebrafish that shows that *GnRH III* knockout has resulted in a male sex-biased population. To date, there is no evidence showing LH and FSH directly regulate the expression of sex-determining genes (*amhr2*, *amhy*, *dmrt1*, *dmy*, *gdf6Y*, *gsdf*, and *sdY*). (**B**) The role of testosterone (T) and oestrogen involved in masculinisation and feminisation in a fish, respectively. Nonetheless, whether the gonads initially differentiated into testes or ovaries followed by sexual differentiation of the brain, or vice versa, is unknown.

**Figure 3 biology-10-00973-f003:**
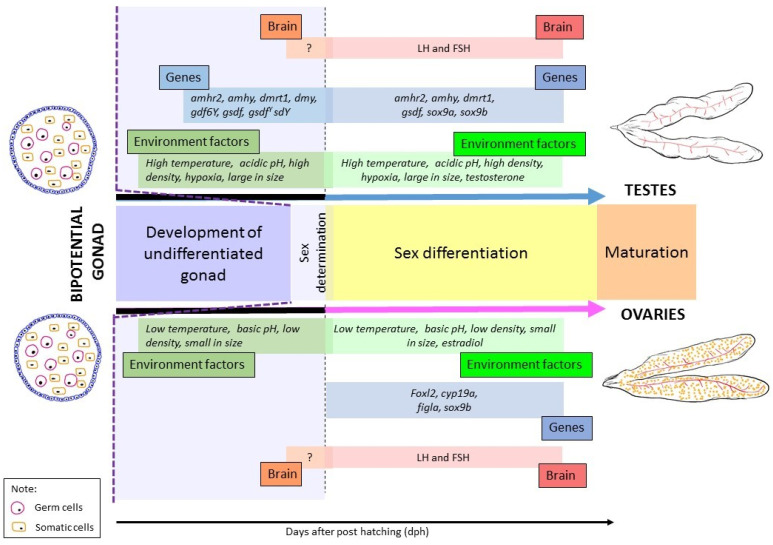
Schematic representation of sex determination/differentiation in teleost. Undifferentiated germ cells need the surrounding somatic cells to provide an instructive signal(s) to initiate sexual differentiation. Therefore, the first step of oogenesis and spermatogenesis is managed by somatic cells [93]. From fish to mammals, Amh signalling plays an important role in gonadal development [13]. In Japanese medaka, *gsdf* null mutants and *amhr2* mutants show excessive growth of germ cells and oocyte arrest during the previtellogenic stage [94]. In zebrafish, *gsdf* and *amh* are essential to inhibit the accumulation of premature oocytes [95]. This suggests the expression of *gsdf* and *amh*/*amhr2* have to be stable during the sex differentiation phase of gonadal development. Reduced *gsdf* or *amh* through the *amhr2* can directly or indirectly result in protandry [53,97,98]. Ovotestis development is the result of differentially expressed *gsdf* gene and in *amhr2* mutants. Therefore, *gsdf* and *amh* signalling is vital for gametogenesis, the production of sex steroids and the secretion of gonadotropins [94].

**Table 1 biology-10-00973-t001:** List of sex-determining genes in teleost species.

Gene Name	Full Gene Name	Chromosome Location	Species	References
*amhr2*	Anti-Müllerian hormone receptor type 2	Autosomal	Grass puffer (*Takifugu rubripes*)	[4]
*amhy*	Y-linked anti-Müllerian hormone	Y chromosome	Nile tilapia (*Oreochromis niloticus*)	[5]
			Cobaltcap silverside (*Hypoatherina tsurugae*)	[33]
			Northern pike (*Esox lucius*)	[34]
			Rockfish (*Sebastes schlegelii*)	[35]
*dmrt1*	Doublesex and mab-3 related transcription factor	Autosomal/Sex chromosome	Spotted scat (*Scatophagus argus*)	[6]
			Chinese tongue sole (*Cynoglossus semilaevis*)	[36]
*dmy*	DM-domain on the Y-chromosome	Y chromosome	Japanese medaka (*Oryzias latipes*)	[7]
*gdf6Y*	Growth differentiation factor 6 on the Y-chromosome	Y chromosome	Turquoise killifish (*Nothobranchius furzeri*)	[8]
*gsdf*	Gonadal soma derived factor	Autosomal	Rainbow trout (*Oncorhynchus mykiss*)	[9]
*gsdf^Y^*	Gonadal soma derived factor on the Y-chromosome	Y chromosome	Philippine medaka (*Oryzias luzonensis*)	[10]
*sdY*	Sexually dimorphic on the Y-chromosome	Y chromosome	Rainbow trout (*Oncorhynchus mykiss*)	[37]
			Atlantic salmon (*Salmon salar*)	[11]
			Brown trout (*Salmo trutta*)	[11]
			Arctic charr (*Salvelinus alpinus*)	[38]

## Data Availability

Not applicable. This review did not report any data.

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
