# Peer review of "Sex Determination and Differentiation in Teleost: Roles of Genetics, Environment, and Brain"

_biology, 2021, doi:10.3390/biology10100973_

Round 1

Reviewer 1 Report

In this manuscript, authors reviewed sex determination and differentiation in Teleost, which is not yet fully known, from the perspective of related genes and the environment. The review paper has been well-organized and systematically written, and I read it very interestingly. Therefore, this review paper will provide researchers who study the sex determination and differentiation study of Teleost with good information on future research directions.

What is regrettable is that there are awkward parts in the expression. So I want to see the English correction.

Author Response

Thank you very much to the reviewer for the comments and feedback. We have performed English editing to this manuscript thoroughly and conscientiously. 

Reviewer 2 Report

Major comments

  1. The authors should carefully read the text and revise some sentences.
  2. line 353: These two references are not cited in References
  3. Once English names with scientific names of fish are used, only the English name should be used thereafter

Minor comments

  1. line 72: ovaries should be ovary
  2. Table 1, lines 101, 103, 107, 108, 112 and throughout the text: Mullerian should be Müllerian
  3. line 96: add Danio rerio after "zebrafish"
  4. line 103: Fallopian should be fallopian
  5. line 110: amhΔ-y?
  6. line 115: delete "fish" after pejerrey
  7. line 134: Is this sentence clear?
  8. lines 95, Table 1, 522: delete "fish" after Medaka (throughout the text) and  Rainbow trout
  9. Table 1: delete "The" in full gene names
  10. line 131: Medaka should be medaka
  11. lines 164, 639: delete (Oncorhynchus mykiss): use rainbow trout therafter
  12. line 226: dissolved?
  13. line 231: add (11-KT) after 11-ketotestosterone
  14. lines 255, 256: use FSH, LH only
  15. line 269: add "in zebrafish"
  16. line 270: add "T" after testosterone
  17. lines 308-310: poor English
  18. line 333: Tilapia should be tilapia
  19. line 387: Prolonged?
  20. line 391: 11-KT but not 11-ketotesotosterone
  21. line 393: delete the scientific name of Nile tilapia
  22. line 396, 397, 550: delete "hormone", "hormones"
  23. line 414: delete and
  24. line 439: delete organism
  25. line 453: Germline Alpha should be germline alpha
  26. lines 476. 477, 478: delete "fish" after medaka, change the scientific name to rainbow trout, Oncorhychus luzonensis should be Oryzias luzonensis?
  27. line 486: add "of" after expression
  28. lines 82, 507: knocked-out and knockout are used, any different meaning?
  29. lines 545, 563: 17β-oestradiol should be E2
  30. lines 616, 617: use only FSH, LH
  31. lines 641, 685, 686, 687: testosterone should be T, oestradiol should be E2
  32. lines 660, 667, 707: use 11-KT, E2
  33. line 735: add The University of Shizuoka, Japan, after Kobayashi
  34. References are horrible, therefore, the  authors need to check whole references carefully: scientific names of animals should be italic, titles should be small capital, journal names should be full name and use Large capital at the first word, for example, General and Comparative Endocrinology, Genome Biology, Environmental Science and Technology, etc.; delete "The" before Journal; add United States of America after National Academy of Sciences,

Author Response

Thank you very much to the reviewer for the comments and feedback. We have done the editing thoroughly based on the reviewers' comments. All the comments have been addressed conscientiously. Our justifications and explanations for the reviewer's comments and feedback are presented in the attachment file.
